# Chronic Variable Stress May Induce Apoptosis in the Testis and Epididymal Sperm of Young Male Rats

**DOI:** 10.3390/biology14060690

**Published:** 2025-06-12

**Authors:** Yeimy Mar De León-Ramírez, Leticia Nicolás-Toledo, Eliut Pérez-Sánchez, Omar Arroyo-Helguera

**Affiliations:** 1Laboratorio de Biomedicina y Salud Pública, Instituto de Salud Pública, Universidad Veracruzana, Av. Luís Castelazo Ayala S/N, Col. Industrial Animas, Xalapa C.P. 91190, Veracruz, Mexico; 2Centro Tlaxcala de Biología de la Conducta, Universidad Autónoma de Tlaxcala, Km 1.5 Carretera Tlaxcala-Puebla S/N, La Loma Xicoténcatl, Tlaxcala C.P. 90070, Tlaxcala, Mexico

**Keywords:** chronic variable stress, oxidative stress, antioxidants, apoptosis

## Abstract

Stressor stimuli are associated with a reduced sperm quality and male infertility, although the effect of stressors on the expression of apoptotic markers in the epididymal sperm and testicles is unclear. This study analyzes the effect of chronic variable stress on the expression of extrinsic and intrinsic apoptotic markers in the testicle and epididymis of exposed rats.

## 1. Introduction

Stress is an unregulated homeostatic state disorder that depends on several adaptive physiological and behavioral responses in the organism [1]. The physiological response depends on the intensity and duration of the stressful stimuli [2]. For example, acute stressors can be single, intermittent, and time-limited, whereas chronic stressors can be sporadic, prolonged, or continuous [3]. In men with high levels of self-reported stress, the relationship between chronic stress and male infertility has been reported. For example, it was found that these men had worse semen parameters than subjects with low levels of self-reported stress [4]. In testis from rats with chronic stress, this induces histological alterations [4], increases oxidative stress, and reduces sperm quality [5].

Chronic stress also reduces the activity of antioxidant enzymes like catalase, superoxide dismutase, glutathione peroxidase, and glutathione in the testis [6]. The imbalance between antioxidant and oxidative stress in the testis could be associated with alterations in sperm quality due to oxidative damage and the activation of cell death by cellular apoptosis. For example, it has been reported that oxidative stress contributes to 30–80% of cases of male infertility by inducing lipid peroxidation and DNA fragmentation by apoptosis [7,8]. Apoptosis is a natural physiological process that occurs during normal animal development and disease states [5]. During spermatogenesis, germ cells and sperm are vulnerable to changes in oxidative stress and therefore are susceptible to oxidation and cellular damage [9], inducing numerous cytopathological changes [10], as well as apoptosis during the maturation of spermatocytes and spermatids [11].

In pathological conditions such as chronic stress, the intrinsic and extrinsic apoptosis pathways in the testis, for example, in diabetes, induce DNA damage in rat testis cells through the p53-p21CIP1/Waf pathway [12]. The expression of p53 promotes proapoptotic responses to induce Bax protein expression in the mitochondria to promote the release of cytochrome C and inhibit the expression of Bcl-2 protein, thereby preventing antagonism to Bax protein action [13]. The p-AKT blocks apoptosis by inhibiting caspase-3 activity [14]. The AP-2α downregulates the Bcl-2 gene and activates the Bax/cytochrome C/caspase-9 in mitochondria to induce apoptosis [15]. The expression of p21 promotes cell cycle arrest in the G1/S phase for DNA repair [16]. However, if the damage is irreversible, it could reinforce the apoptotic signal of p53 to induce the activation of Bax protein and the apoptosome caspase-9, and finally the activation of executioner caspases such as caspase-3 [13]. In addition, the extrinsic death receptor-mediated pathway can also induce cell death in testicular germ cells and Sertoli cells [17].

In the testicle, the activation of the FAS receptor by its ligand (FASL) leads to the activation of caspase-8, which activates caspase-3 and triggers apoptosis [18]. The inhibition of other regulators, such as PPAR-γ, reduces lipid peroxidation and oxidative DNA damage [19]. Under hypoxia conditions in the testis, HIF-1α potentiates apoptosis to induce p53 in the varicocele [20]. After being produced in the testis, sperm travel to the epididymis, where germ cells mature and capacitation takes place [21]. In the epididymis, the administration of tributyl hydroperoxide as a stressor induces DNA fragmentation and oxidative stress associated with sperm quality alterations [22]. However, the variable effects of the apoptotic pathways involved in chronic stress have not yet been studied. The goal of this study was to analyze the expression of intrinsic and extrinsic markers of apoptosis in the testis and epididymal sperm extracts of rats exposed to chronic variable stress.

## 2. Materials and Methods

### 2.1. Experimental Design

Male Wistar rats weaned at postnatal day 21 (60–80 g) were housed individually and maintained under a 12:12 h light/dark cycle with a room temperature of 20 ± 2 °C. The rats were randomly assigned to experimental groups: control *(n* = 6–7) and stress group (*n* = 6–7). The control and stress groups had free access to tap water and a standard rodent diet (Purina Laboratory Chow 5001). The stress animal group was exposed to a chronic variable stress model for 4 weeks, except for the weekends. After four weeks, all animals were deeply anesthetized using Nembutal (33 mg/kg b.w.) and decapitated using a rodent guillotine device (Harvard-Apparatus, Holliston, MA, USA). The Research Ethics Committee from the Universidad Autónoma de Tlaxcala approved all the experimental procedures according to Mexican Guidelines for Animal Care, which are based on recommendations by The Association for Assessment and Accreditation of Laboratory Animal Care International (Norma Oficial Mexicana NOM-062-Z00-1999).

### 2.2. Chronic Variable Stress Procedure Model

The rats were subjected to a series of alternative and random individual stressor stimuli each day at different times. The following stressors were used: (a) Space reduction in clean box stimuli, in which the seven rats were placed in a collective clean box designed for four to five rats, with dimensions of 50 cm × 40 cm × 21 cm, for 5 h; (b) space reduction in dirty box stimuli, in which the seven rats were placed in a collective dirty box designed for four to five rats, with dimensions of 50 cm × 40 cm × 21 cm, for 5 h; (c) restraint stimuli, for 3 h, in which each rat, one by one, was placed into a plastic tube of 6 cm × 20 cm, avoiding movement and risk of tipping over; (d) forced swimming stimuli in cold water, in which each rat, one by one, was placed together in a plastic tank with dimensions of 100 cm height × 50 cm diameter and containing 75 cm of cold water at 18 °C for 10 min; and (e) forced swimming stimuli in warm water (same procedure before at 28 °C for 10 min. Animals remained on their diets throughout stress exposure. Stress application started at different times every day to minimize its predictability [23,24].

### 2.3. Epididymal and Complete Testicles Extract

The rat was placed in a supine position and a longitudinal incision was made along the ventral midline of the abdominal wall to extract the complete testicles and weigh them on a scale (ACCURIS W3300-500). Both testicles were frozen in an ultra-freezer at −80 °C for subsequent analysis. The epididymal sperm extract was taken from the middle to the cauda of the epididymis, with a single incision, and placed in an Eppendorf tube containing 2 mL of phosphate-buffered saline (PBS). A tissue sample from the cauda epididymis (300 mg) or complete testicle was taken and placed in an Eppendorf tube containing 2 mL of phosphate-buffered saline (PBS) previously heated and maintained in a water bath at 37 °C. Once in the tube, the sample was macerated using a scalpel blade to facilitate sperm diffusion within the solution, and 10 min later, several aliquot samples of 10 µL were taken and stored at −20 °C for future measures. The complete testicle or sperm epididymal extracts were homogenized in lysis buffer (50 mM Tris-HCl, pH 7.4, 0.2% sodium deoxycholate, 0.2% sodium dodecyl sulfate (SDS), 1% Triton X-100, 1 mM sodium ethylenediaminetetraacetate, 1 mM phenylmethylsulphonyl fluoride, 5 µg/mL of aprotinin, 5 µg/mL of leupeptin) with a silent crusher. Cell debris was removed by centrifugation, and the protein concentration was determined by the standard Bradford protein assay.

### 2.4. Oxidative Stress and Total Antioxidant Status

The concentration of the lipid peroxidation product malondialdehyde (MDA) was determined spectrophotometrically. Protein extracts were mixed with TRIS buffer (100 mM, pH 7.5), 20% acetic acid pH 3.0, and 0.4% Thiobarbituric acid (TBA). In a thermoblot, all samples were warmed to 100 °C for 45 min and cooled in ice, and 1.2% of KCl was added to stop the reaction. Then, the overnadant was measured at 532 nm using a microplate reader (Spectramax Plus; Molecular Devices, Sunnyvale, CA, USA) and the data were expressed in nmol MDA/mg protein. To determine the Total Antioxidant Status (TAS), the sample, blank H_2_O or ascorbic acid standard, and FRP buffer (25 mL of 300 mmol/L sodium acetate pH 3.6, 2.5 mL of 50 mmol/L K3Fe (CN) 6, and 2.5 mL of 20 mmol/L FeCl_3_ 6H_2_O) were incubated for 3 min at 37 °C. Absorbance was read at 593 nm using a microplate reader (Spectramax Plus; Molecular Devices, Sunnyvale, CA, USA). The TAS was determined by interpolation with a seventh-point calibration curve of known amounts of ascorbic acid. The data are expressed as millimoles of ascorbic acid equivalents/mg protein.

### 2.5. Determination of mRNA by Quantitative RT-PCR

RNA extraction was performed with TriReagent (Sigma-Aldrich, St. Louis, MO, USA). The epididymal extract, TriReagent, and chloroform were incubated at room temperature in 1.5 mL Eppendorf tubes. Then, it was manually mixed and incubated for 10 min at room temperature, and centrifuged at 12,000 rpm for 10 min at 4 °C. The aqueous phase was transferred to another Eppendorf tube and an isopropanol was added. After incubation, the mixture was centrifuged at 12,000 rpm for 10 min at 4 °C. The pellet was washed 3 times with 70% cold ethanol. The remaining ethanol evaporated, and the pellet was resuspended in sterile water. The RNA concentration was calculated with a Nanodrop spectrophotometer (ND-1000 Nanodrop Technologies, Inc., Wilmington, DE, USA) at an absorbance of 260–280 nm, and the RNA integrity was analyzed by horizontal electrophoresis in a 1% agarose gel, stained with SYBR Green fluorescence marker. For cDNA synthesis, 100–200 ng of the total RNA was reverse transcribed using ReverTra Ace^®^ qPCR RT master mix, and quantitative real-time PCR was performed with a SensiFAST™ SYBER^®^ No-ROX Kit (SIGMA) in a Piko Real-Time PCR System (Thermo Sci, Boston, MA, USA). Table 1 shows the primers used to analyze the PCNA-1, CREB, p21, PPAR-γ, p53, Bax, Bcl-2, and HIF-1α expression. The denaturation step was carried out at 94 °C for 5 min, and the next step was performed via 30 cycles of 35 s at 94 °C, 30 s at 60 °C, and 30 s at 72 °C, followed by a final step at 22 °C. The relative expression was calculated according to the housekeeping gene β-actin normalized as an internal control with the 2^−ΔΔCT^ method.

### 2.6. Western Immunoblotting

Frozen epididymal sperm extract or total testicle samples were disrupted in 10% wt/vol in lysis buffer (10% wt/vol) [50 mM Tris-HCl, pH 7.4, 1% Triton X-100, 0.2% sodium deoxycholate, 0.2% sodium dodecyl sulfate (SDS), 1 mM sodium ethylenediaminetetraacetate, 1 mM phenylmethylsulphonyl fluoride, 5 µg/mL of aprotinin, 5 µg/mL of leupeptin] with a silent crusher. Tissue and cell debris were removed by centrifugation at 12,000 rpm at 4 °C for 10 min, and the protein concentration was measured in the supernatant by the Bradford method. Lysates of protein (10 μg) under reducing conditions were resolved in SDS-polyacrylamide gel (12%), transferred onto the nitrocellulose membrane overnight, and blocked for 2 h at room temperature with 5% fat-free milk in tris-buffered saline (TBS). After 12 h of incubation at 4 °C with the primary antibody (β-Actin antibody 1:2000 dilution; PPAR-γ antibody 1:1000 dilution; AP-2α antibody 1:1500 dilution; FAS antibody 1:1500 dilution; C/EBP-β antibody 1:1000 dilution; P-Akt antibody 1:1500 dilution (all of them from Santa Cruz Biotechnology, Inc., USA), membranes were washed thrice for 10 min with TBST (tris-buffered saline pH = 7.4 with 0.1% tween-20) and incubated with a specific secondary antibody [goat anti-mouse IgG AP 1:1000 dilution (Santa Cruz Biotechnology, Inc., USA) and goat anti-Rabbit IgG AP 1:1000 dilution (PROMEGA, Madison, WI., USA)] diluted in TBST with BSA at 10% for 2 h at room temperature. Then, the membrane was washed with TBST (15 min each). The intensity of the immunoblotting bands was measured using BCIP^®^/NBT solution (B6404-Sigma Aldrich, St. Louis, MO, USA). Moreover, chemiluminescence (ECL) was used to detect signals under the gel doc apparatus (Bio-Rad Laboratories, Inc. Ciudad de México, México). The immunoreactive bands were analyzed by Image Lab Software 6.0.1. (Bio-Rad Laboratories, Inc.). β-actin was used as a loading control. The intensity of each immunoblotting band of the proteins of interest was divided by the intensity of the β-actin band, and the results were expressed in arbitrary units (A.U.).

### 2.7. Caspase-3 Assay

According to the method described by Sigma-Aldrich, the enzymatic activity of caspase-3 was assessed using a colorimetric assay. The testis tissues and epididymal sperm extracts were homogenized 1:10, *w*/*v*, in lysis buffer and protease inhibitors. In a flat-bottom 96-well plate, 5 μL of cell lysate or Caspase-3 positive control was added. Then, 85 μL of 1× Assay Buffer and caspase-3 inhibitors were added to all the wells. The reaction was started by adding 10 μL of caspase-3 substrate to each well, covering the plate, and mixing gently for 5 min at room temperature at 900 RPM, always trying to avoid the formation of bubbles in the wells. This was incubated at 37 °C for 90 min and the absorbance was read at 405 nm.

### 2.8. Statistical Analysis

The results were expressed as mean ± standard error (SE). The Shapiro–Wilk test was used to analyze the normality of the data distribution. Student *t*-tests were used to determine significant differences between the control and stressed groups, and significant differences were considered at *p* ≤ 0.05. GraphPad Prism v. 9 was used for statistical analyses (GraphPad Software, San Diego, CA, USA).

## 3. Results

### 3.1. Oxidative Stress, Antioxidant Status, and Apoptotic Marker’s Expression in the Complete Testicle from Rats Exposed to CVS

For this analysis, a complete testicle extract was used to measure oxidative stress as the TBARS content [*t* (1,14) = 2.110; *p = 0.0533*] and total antioxidant status [*t* (1,14) = 0.7560; *p* = 0.4622] were not statistically significant in the CVS group compared to the control group (see Figure 1).

The phosphorylated p-Akt and AP-2α protein levels decreased, while FAS increased; these expression changes were significant in the CVS group compared with the control group [*t* (1,10) = 3.100, *p* = *0.0078*; *t* (1,10) = 8.847, *p* < *0.0001*; *t* (1,10) = 2.549, *p = 0.0214*; respectively; see Figure 2B]. The quantification of the PPAR-γ and C/EBP-β protein levels increased; these changes were significant between the control and CVS group [*t* (1,10) = 1.935, *p* = 0.0379], [*t* (1,10) = 5.278, *p* = 0.0003], respectively, as shown in Figure 2C. The caspase-3 activity was significantly higher in complete testicles in the CVS group compared to the control group [*t* (1,14) = 2.984, *p* < *0.01*].

### 3.2. Oxidative Stress, Antioxidant Status, and Apoptosis Markers Expression in the Epididymal Sperm Extract from Rats Exposed to CVS

The oxidative stress status shows a significant difference between the control and CVS groups [*t* (1,14) = 2.349; *p = 0.0340*]. Likewise, significant differences were shown in the total antioxidant status [*t* (1,14) = 3.457; *p = 0.0038*], as shown in Figure 3. The sperm quality was reported previously in unpublished data from the master thesis of Nolasco Garduño [24] and was corroborated in this study. All the parameters decreased in the CVS group, with increased motility (*t* = 2.237, *p* = 0.0378), viability (*t* = 2.569, *p* = 0.0249), concentration (*t* = 2.613, *p* = 0.0225), and morphology (*t* = 7.54, *p* = 0.0001) compared with the control group.

As the levels of phosphorylate p-Akt protein decrease, the protein levels of AP-2α decrease, and FAS increases; both changes were significant between the control and CVS group [*t* (1,10) = 10.81, *p = <0.0001*; *t* (1,10) = 17.32, *p* < *0.0001*; *t* (1,10) = 3.927, *p = 0.0015*; respectively]. In Figure 4B,C, it is shown that the expressions of PPAR-γ and C/EBP-β were higher in the epididymal sperm extract of the CVS group compared to the control group [*t* (1,10) = 2.984, *p = 0.0124*] and C/EBP-β [*t* (1,10) = 2.638, *p = 0.0195*], respectively. The caspase-3 activity increased significantly in the epididymal sperm extracts of the CVS group [*t* (1,14) = 2.188, *p* < *0.05*] vs. the control group, as shown in Figure 4C.

### 3.3. Effect of CVS on the Expression of Apoptosis Markers in Epididymal Sperm Extract

Figure 5 shows changes in the expression of apoptotic markers after CVS in the epididymal sperm extract. The results show a significant increase in the proapoptotic markers p53 [*t* (1,10) = 14.77, *p = 0.0001*], p21 [*t* (1,10) = 6.288, *p = 0.0001*], PPAR-γ [*t* (1,10) = 7.093, *p = 0.0001*], HIF-1α [*t* (1,10) = 7.383, *p* < *0.0001*], and Bax [*t* (1,10) = 4.102, *p = 0.0005*]. Nevertheless, there is a reduction in the mRNA expression of PCNA-1 [*t* (1,10) = 3.571, *p = 0.0017*], Bcl-2 [*t* (1,10) = 4.211, *p = 0.0004*], and CREB [*t* (1,10) = 3.549, *p = 0.0018*].

## 4. Discussion

The interaction between ROS and antioxidant enzymes is well documented; however, the role they play in male reproductive organs and their greater impact on fertility remains unclear. In this study, contrasting data related to the antioxidant and oxidative state between the whole testicle and the epididymal sperm extracts of rats exposed to CVS were obtained, observing a decrease in oxidative stress levels in the testis with no changes in the overall antioxidant status; meanwhile, in the epididymal sperm extract, oxidative stress and the antioxidant status increased. These changes are likely to respond to hormonal changes and to the enzymes involved in the antioxidant state that protect the reproductive system. Regarding the involvement of hormones, it has been reported that chronic variable stress decreased serum testosterone levels, increased ROS, and caused damage to the membrane lipids of the testicular cells [11]. These suggest that the balance of glucocorticoids and androgens could be related to the antioxidant/oxidant status in the testis. In addition, this study shows a clear trend of 50% lower MDA levels in the CVS group than in the control group. In other studies, it has been reported that stress stimuli could increase the levels of dihydrotestosterone (DHT), a biologically active metabolite of the hormone testosterone; it is formed in the prostate gland and testicles by the enzyme 5α-reductase, this hormone increasing due to exercise-related stress [30]. Also, previous studies have demonstrated the antioxidant power of DHT in response to allostatic changes and adaptative mechanisms to mitigate the damage induced by stress stimuli and to prevent the formation of lipid peroxides (LPO) [31].

The epididymis is equipped with a battery of antioxidant enzymes that maintain the ROS at physiological levels to achieve the goal of protecting the maturing spermatozoa. This antioxidant/oxidative stress balance can be altered by stress stimulus such as tert-butyl hydroperoxide, which increased the enzyme activities of PRDX1 and PRDX6 but not SOD in spermatozoa from the cauda epididymis, to protect the maturing spermatozoa from high levels of ROS [32]. In this study, the activities of PRDX1 and PRDX6 were not measured; however, a decrease in the p-Akt levels was found in the rats exposed to CVS. Previously, it was reported that the inhibition of Akt phosphorylation by lysophosphatidic acid, a product of PRDX-6, increases the O_2_^−^, ROS, and lipoperoxidation associated with the loss of viability and high DNA fragmentation in spermatozoa from the epididymis [33,34]. In addition to that, C/EBP-β regulates the expression of the porcine Prdx6 gene at both the mRNA and protein level [35]. In this study, we found that C/EBP-β and p-Akt increased in the CVS group, suggesting that PRDX6 could participate in the antioxidant effects caused by CVS in the testicle and epididymis. In the epididymal sperm extracts, CVS increased the oxidative stress and antioxidant status, suggesting that the overexpression of the antioxidant status diminishes the damage to oxidative stress in the epididymal sperm. Different studies report that stress stimuli lead to decreased SOD or CAT activities and total GSH [11]. In this study, the contrasting data in the complete testicle extract related to the decreased oxidative stress caused by CVS may have been masked by other cellular components such as leukocytes and Sertoli cells; in this regard, the stress induced with retinol on Sertoli culture significantly increased TBARS, conjugated dienes, and hydroperoxides [36], or the TBARS increased due to the sodium fluoride-induced oxidative stress on the testicle [37]. Therefore, it is necessary to corroborate these findings and determine the hormonal involvement and antioxidant defense of CVS-induced apoptosis in reproductive tissue.

Apoptosis in reproductive tissues is a crucial physiological process that ensures the removal of defective sperm cells and maintains the quality of the sperm population; in addition, serine/threonine-specific protein kinase (p-Akt) plays a significant role in cell survival pathways by inhibiting apoptotic proteins, as Bax [38] and low concentrations of p-Akt are associated with increased apoptosis, and the phosphorylation of Akt is regulated by PRDX6 in sperm cells [34]. Our results show that CVS reduced p-Akt and AP-2α protein levels in the complete testicle and epididymis extracts to enhance the activation of the apoptotic pathway. AP-2α is a transcription factor that downregulates Bcl-2, an anti-apoptotic gene [15]. In this study, CVS increased the expression of FAS in the testis and epididymal sperm; this transcription factor is a well-known death receptor that mediates apoptosis through the extrinsic pathway, and the binding of the FAS ligand to FAS triggers the formation of the death-inducing signaling complex, leading to caspase activation and cell death in various contexts, including reproductive health [17].

In addition, increased PPAR-γ expression was found in the testis and epididymal sperm extracts of the CVS group. Previous studies have shown that this nuclear receptor regulates lipid metabolism, glucose homeostasis, cell differentiation, and apoptosis in various cell types [39]. In the context of sperm cells, elevated PPAR-γ may promote apoptosis by influencing the expression of genes involved in the apoptotic pathway. For instance, PPAR-γ activation can lead to the upregulation of pro-apoptotic proteins such as Bax and the downregulation of anti-apoptotic proteins like Bcl-2 [19]. Another factor that has been implicated in the stress-induced inflammatory process is C/EBP-β, a transcription factor that plays a role in regulating cellular proliferation, differentiation, and apoptosis, and high levels of C/EBP-β are associated with increased apoptosis by modulating the expression of genes involved in the apoptotic pathway. In sperm, C/EBP-β may enhance the transcription of pro-apoptotic genes, contributing to increased apoptosis [40].

The apoptotic intrinsic pathway of apoptosis is mediated by p53, a tumor suppressor marker that induces cell cycle arrest and apoptosis in response to cellular stress. In this study, high levels of p53 and p21 expression were found in the CVS group. In this way, high p53 mRNA expression has been reported to enhance the transcription of pro-apoptotic genes, promoting apoptosis in the mouse spermatogonia and spermatocyte germ cell lines exposed to higher rates of electromagnetic field and inducing p53/p21-mediated cell cycle arrest and apoptosis [16]. Nevertheless, concerning PCNA-1, our results show decreased levels due to the CVS. PCNA and AP-2α are essential for DNA replication and repair [41] and play a crucial role in cell proliferation. These low levels of mRNA expression in PCNA-1 have been associated with reduced cell proliferation, contributing to increased apoptosis in spermatogenic cells. Studies have shown that decreased PCNA expression correlates with elevated sperm apoptosis rates, suggesting its importance in maintaining spermatogenic cell survival [41]. Another transcription factor that is important in our results is HIF-1α, a transcription factor that is activated under hypoxic conditions or oxidative stress mediated by oxygen radicals. In this study, we observed the high mRNA expression of HIF-1α, consistent with an oxidant/oxidative stress imbalance in the epididymal sperm extract. In accord, HIF-1α is associated with increased apoptosis under hypoxic conditions, which affects spermatogenic cells and leads to impaired spermatogenesis [42]. The expression levels of Bcl-2 and Bax are related to intrinsic and extrinsic apoptotic pathways, and in this study, Bcl-2 decreased while the expression of Bax mRNA levels increased only in the CVS group. In accordance, this correlates with increased apoptosis in germ cells in the testicles, affecting sperm production [16]; in response to heat stress, the germ cells are regulated by the Bcl-2 and Bax expression associated with apoptosis [43]. Finally, low levels of CREB were found in the epididymal sperm extract in the CVS group. CREB is a transcription factor involved in cell survival and differentiation, and low mRNA expression results in the impaired transcription of survival genes, facilitating apoptosis. In germ cells, reduced CREB expression and increased apoptosis has been reported, impacting spermatogenesis [44].

## 5. Conclusions

According to our results, it could be concluded that CVS damage triggers the induction of apoptosis markers by intrinsic (PPAR-γ, p53, p21, HIF-α, and Bax) and extrinsic (p-Akt, AP-2α, and FAS) caspase-3-dependent pathways in both complete testicles and epididymal sperm extracts. This study supports the view that the stressor stimulus could be involved in the infertility process, and future studies should identify the role of the antioxidant and hormonal status in apoptotic gene expression.

## Figures and Tables

**Figure 1 biology-14-00690-f001:**
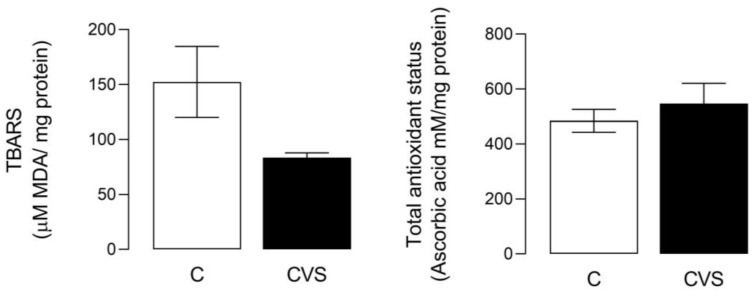
Oxidative stress and total antioxidant status in complete testicles from the CVS group. TBARS (Thio barbituric acid reactive substances) was measured as an oxidative stress marker and the total antioxidant status was measured as FRP, as shown in the methodological section. *t*-test was performed; data are represented as mean ± SE (n = 7 per group). Chronic variable stress (CVS).

**Figure 2 biology-14-00690-f002:**
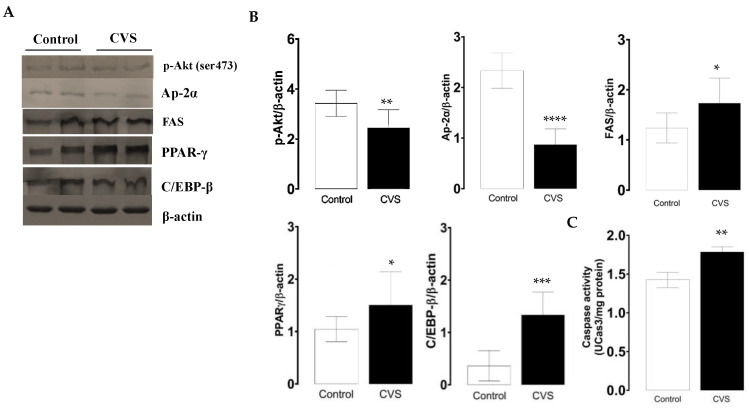
Expression of molecular markers from the extrinsic apoptotic pathway and caspase-3 activity in the complete testicle of rats exposed to CVS. Rats were euthanized after CVS. (**A**) 200 μg of protein from testis cell lysates (control and CVS groups) was subjected to SDS-PAGE electrophoresis and immunoblot analysis, as shown in the methodology section. (**B**) Densitometric quantification was normalized to the β-actin levels (See Appendix A). (**C**) Caspase-3 activity. The *t*-test was performed, and bars represent mean ± SE, n = 6 for all groups. * *p* < *0.05*, ** *p* < *0.01*, *** *p* < *0.001*, **** *p* < *0.0001* vs. control group. Chronic variable stress (CVS).

**Figure 3 biology-14-00690-f003:**
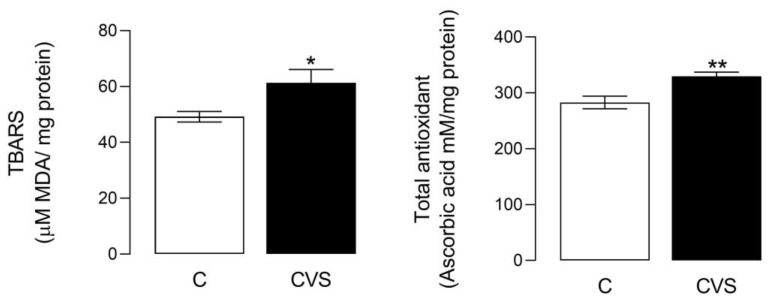
Oxidative stress and total antioxidant status in the epididymal sperm extracts of rats exposed to CVS. Thiobarbituric acid reactive substances (TBARS) were measured as an oxidative stress marker, and the total antioxidant status was measured as FRP, as shown in the methodological section. A *t*-test was performed; data are represented as mean ± SE (n = 7 per group). Chronic variable stress (CVS). * *p* = 0.05; ** *p* = 0.001 vs. control group.

**Figure 4 biology-14-00690-f004:**
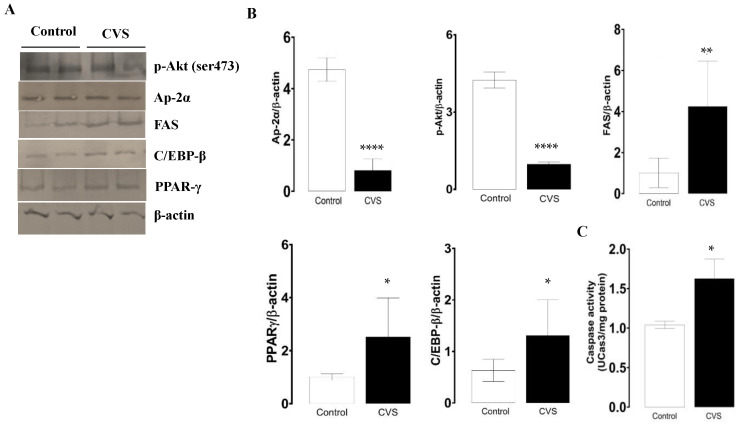
Expression of molecular markers from the extrinsic apoptotic pathway and caspase-3 activity in the epididymal sperm extracts of rats exposed to CVS. Rats were euthanized after CVS. (**A**) 200 μg of protein from epididymal sperm extracts (control and CVS groups) was subjected to SDS-PAGE electrophoresis and immunoblot analysis, as shown in the methodology section. (**B**) Densitometric quantification was normalized to the β-actin levels (see Appendix A). (**C**) Caspase-3 activity. The *t*-test was performed, and bars represent mean ± SE, n = 6 for all groups. * *p* < *0.05*, ** *p* < *0.01*, **** *p* < *0.0001* vs. control group. Chronic variable stress (CVS).

**Figure 5 biology-14-00690-f005:**
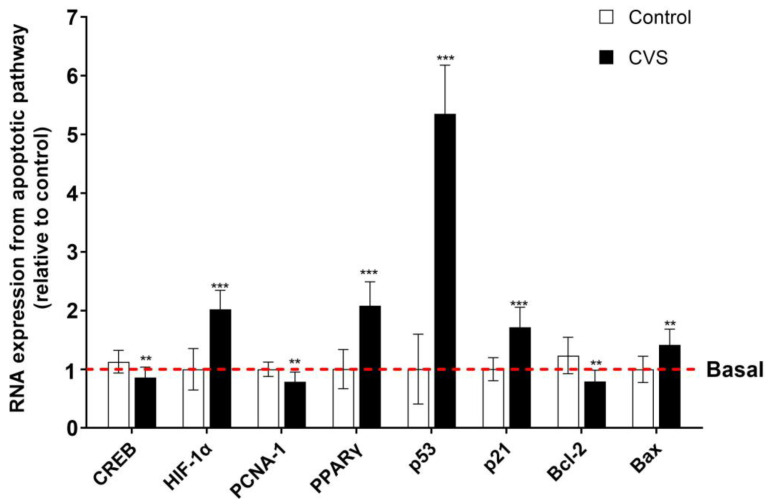
Expression of apoptotic markers in the epididymal sperm extract of rats exposed to CVS. Total RNA was extracted, and the relative mRNA expressions of p53, p21, PCNA-1, PPAR-γ, HIF-1α, Bax, Bcl-2, and CREB were detected by quantitative RT-PCR. A *t*-test was performed, and bars represent mean ± SE values by duplicate determination. ** *p* < *0.01*, *** *p* < *0.001* vs. control group (n = 6). Chronic variable stress (CVS).

**Table 1 biology-14-00690-t001:** RT-PCR Primer sequences.

Gene	PMID/Reference	Sequence 5′ at 3′	Size (pb)	AT (°C)
β-actin	30625859 [25]	F: AGCCATGTACGTAGCCATCC R: CTCTCAGCTGTGGTGGTGAA	228	55
PCNA-1	31934015 [26]	F: GGAGACAGTGGAGTGGCTTT R: AGTTTTCTGCGAGTGGGGAG	1287	55
CREB	30625859 [25]	F: CCTCTCTCTTTCGTGCTGCT R: GGCCTGCAGACATTAACCAT	256	55
Bcl-2	29085599 [27]	F: CCAGGAGAAATCAAACAGAG R: GTGGATGACTGAGTACCT	118	62
Bax	29085599 [27]	F: CCAGTTCATCTCCAATTCG R: CTACAGGGTTTCATCCAG	133	62
p21	29085599 [27]	F: CTTGGAGTGATAGAAATCTGTCA R: CTTGCACTCTGGTGTCTG	107	62
PPAR-y	17461532 [28]	F: GTCCTCCAGCTGTTCGCCA R: TGATATCGACCAGCTGAACC	793	58
HIF-1α	12824304 [29]	F: CCACCTCTTTTTGCAAGCAT7 R: AAGAAACCGCCTATGACGTG	301	60
p53	29085599 [27]	F: TGGGCATCCTTTAACTCTA R: GTATTTCACCCTCAAGATCC	84	62

## Data Availability

All data are contained within the article or Appendix A.

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
