# Peer review of "Chronic Variable Stress May Induce Apoptosis in the Testis and Epididymal Sperm of Young Male Rats"

_biology, 2025, doi:10.3390/biology14060690_

Round 1

Reviewer 1 Report

Comments and Suggestions for Authors

1. In conclusion the authors have mentioned that there is no role of oxidative stress in the apoptosis and CVS independently affects the apoptosis, However, the authors have linked CVS with oxidative stress in the figure placed after the abstract to summarize their findings.

2. How many sperm were used in each experiment or replicate?

3. How the authors determined the number of sperms?

4. If the number of sperm were not determined in each experiment or replicate, then how can the authors answer the differential expression of mRNA and proteins due to imbalanced sperm numbers?

5. Sperm parameters should be included in the study to elaborate the effect of induced stress on morphology and quality of sperm (concentration, motility etc.)

6. Figure 2: use similar expression trends for significance i.e., bar with asterisks. Make all the figures similar regarding pattern and expressions i.e., box outline in E is different from others.

7. Figure 2C: the control groups show big difference in the intensity of protein bands for both PPAR-γ and C/EBP-β. Why are intensities different in the same group? Please provide the original gel diagrams.

8. Please mention the primers used in this study in tabular form to help others to repeat or use these primers with convenience. Also provide information about melting point, size and reference of used primers either from published papers or online database.

9. Please replace 2−ΔΔCT with 2−ΔΔCT.

10. Please replace RNAm with mRNA in abstract.

11. Introduction should be divided into different paragraphs according to scientific requirements.

Author Response

Dear Editor and reviewer´s

Thank you very much for taking the time to review this manuscript. Please find the detailed responses below and the corresponding corrections highlighted in red in the re-submitted file:

Reviewer 1

  • Observations 1: In conclusion the authors have mentioned that there is no role of oxidative stress in the apoptosis and CVS independently affects the apoptosis, However, the authors have linked CVS with oxidative stress in the figure placed after the abstract to summarize their findings.

Answer: Thank you for the observation, the conclusion was corrected and is consistent with the results obtained in this manuscript and consistent with the abstract figure. This supported in the figure 1 where the content of MDA was not significant statistically (p value was p=0.0533) but it is showing a clear trend to decrease in a chronic CVS group in the testes and was significant in epididymis sperm. In accord with previous reports and our data we linked CVS with oxidative stress as possible stimuli of apoptosis in this study and elaborate the figure abstract.

  • Observations How many sperm were used in each experiment or replicate? 3. How the authors determined the number of sperms? 4. If the number of sperm were not determined in each experiment or replicate, then how can the authors answer the differential expression of mRNA and proteins due to imbalanced sperm numbers?

Answer: Thank you for the observation, In this article, sperm quality is not reported because it was published previously in the thesis of Nolasco Garduño, which can be consulted in this link:  Universidad Autónoma de Tlaxcala: Efecto del estrés crónico variable sobre el arreglo histológico e inflamación testicular y estrés oxidativo espermático epididimal en ratas jóvenes adultas) link   http://repositorio.uatx.mx:8443/jspui/handle/DSyTI_UATx/201. It is important to mention that we use the same methodology and the biological samples in each experiment and made them twice in our study. Also, in the section methodology we report that 7 rats were used for each group, and we had 7 samples of testis and epididymis sperm, also we made a double measure of each component.

 In addition, we used total RNA, previously quantified by spectrometry, 1 ug was used for RT and then the mRNA expression for each gene were normalized with the housekeeping gene actin. The protein levels were quantified by Bradford method, and we use B-actin to normalized the protein expression in the immunoblot assay. This data was added in the methodology section.

  • Observations 5. Sperm parameters should be included in the study to elaborate the effect of induced stress on morphology and quality of sperm (concentration, motility etc.)

Answer: Thank you for the observation: The results that the revisor wants are already determined and published before in the master thesis from Vanessa Guadalupe Nolasco Garduño link:  http://repositorio.uatx.mx:8443/jspui/handle/DSyTI_UATx/201, we added the information in the methodology and discussion section.

  • Observations 7: Figure 2C: the control groups show big difference in the intensity of protein bands for both PPAR-γ and C/EBP-β. Why are intensities different in the same group? Please provide the original gel diagrams.

Answer: Thank you for the observations: the original gel diagrams are included in the complementary documents that we have already sent.

  • Observation 8. Please mention the primers used in this study in tabular form to help others to repeat or use these primers with convenience. Also provide information about melting point, size and reference of used primers either from published papers or online database.

Answer: Thank you for the observation. The table 1 was add in the article.

  • Observation 9, Please replace 2−ΔΔCT with 2ΔΔCT and Please replace RNAm with mRNA in abstract.

Answer: Thank you for the observation: the corrections were done

  • Observation 11. Introduction should be divided into different paragraphs according to scientific requirements.

Answer: Thank you for the observation: the corrections were done

Reviewer 2 Report

Comments and Suggestions for Authors

In the submitted MS, authors present the results of a comparative biochemical and molecular analysis of epididymal and testis extracts from control rats and those after chronic stress. They conclude that chronic variable stress leads to increased sperm apoptosis regardless of the antioxidant status of the organ. Given the statistics of male infertility, new data on the relationship between stress and sperm quality are of potential interest to a wide range of specialists. However, the MS requires significant revision before publication.

Major points

  1. Authors demonstrate a change in the expression level of some apoptotic factors and, based on this, conclude that apoptosis processes are enhanced. I suppose that for final verification of this conclusion, authors should also show a change in the number of apoptotic cells, for example, using annexin V or other markers.
  2. Authors confidently assert that all the changes they described occur in spermatozoa. Indeed, when the epididymis is ground, motile spermatozoa will be the first to enter the medium in significant quantities. However, contamination of this suspension with other cells is possible. The likelihood of such contamination is especially high when preparing testicular specimens, since the spermatozoa in the testicular tubules are not yet capable of movement. What tests did authors perform to exclude contamination of the samples with cells other than spermatozoa?
  3. Authors should describe the Chronic Variable Stress procedure in more detail. They claim to have used “one different stressor per day.” How was the order of application of different stressors determined? Were all animals in the experimental group exposed to the same stressor on the same day? How were 8 rats (p. 3) placed in one cage if n = 6-7? When five rats were placed together in a plastic tank, how were these animals selected from a group of 7 animals? Finally, authors should explain why they used the Variable Stress procedure rather than chronic exposure to a single stressor, such as restraint by a plastic tube.
  4. The authors' data on the content of malondialdehyde in the testicles and epididymis seem worthy of attention (Figs. 1 and 3). The mean for the testes, as can be seen in Fig. 1, are almost twice as high for the control group as for the experimental group. How can authors explain the reduction in oxidative stress during chronic stress exposure? In addition, the standard error (SE) in these two groups are very small and do not even overlap. Are the differences between these groups really not significant? The much smaller differences in mean values in the case of the epididymis are, in the authors' opinion, reliable. It is also worth noting that the average levels of malondialdehyde in the epididymis of control rats are significantly lower than in the testes. If so, what features of the biochemistry of these organs might be associated with this pattern?

Minor point

The MS is formatted with errors. In particular, commas and periods are missing in many places in the references in the text. The reference Wang et al., 1997 (p. 10) is missing from the list of references.

Author Response

Dear Editor and reviewer´s

Thank you very much for taking the time to review this manuscript.

Round 2

Reviewer 1 Report

Comments and Suggestions for Authors

The sperm parameters should have been included as observations vary from researcher to researcher.

Moreover, the provided gel diagrams do not contain their own marker proteins and the markers are attached separately with the diagrams. It would be better if each diagram contains its own marker to make them unsuspicious.

Please improve the conclusion part with more details.

Author Response

Dear Editor and reviewer´s

Thank you very much for taking the time to review this manuscript. Please find the detailed responses below and the corresponding corrections highlighted in red in the re-submitted file:

Reviewer 1

  • Observations 1: The sperm parameters should have been included as observations vary from researcher to researcher.

Answer: thank you for the observation, was add in the result section and discussion the follow test:

Results (258-262 lines): The sperm quality was reported previously in unpublished data from the master thesis of Nolasco Garduño [24] and was corroborated in this study. All the parameters decreased in the CVS group with progressive motility (t= 2.237, p=0.0378), viability (t=2.569, p=0.0249), concentration (t=2.613, p=0.0225), and morphology (t= 7.54, p=0.0001) compared with the control group.

Discussion (342-344 lines): In this study, CVS triggers the induction of apoptosis markers involved in the intrinsic and extrinsic pathways; this is consistent with the decrease in viability and motility in the sperm of the CVS group, previously reported by Nolasco Garduño [24

  • Observations Moreover, the provided gel diagrams do not contain their own marker proteins, and the markers are attached separately with the diagrams. It would be better if each diagram contains its own marker to make them unsuspicious.

  • Answer: Thank you for the observation. A complete gel diagrams were added with own marker proteins as you see in the supplementary material; Figure S1.

  • Answer: Observations 5. Please improve the conclusion part with more details.

Answer: Thank you for the observation. The conclusion was modified in both sections as follows: 

Abstract:    (29-32 lines)  In conclusion, CVS damage triggers the induction of apoptosis markers by intrinsic (PPAR-γ, p53, p21, HIF-α, and Bax) and extrinsic (p-Akt, AP-2α, and FAS) caspase-3 dependent pathways in both testicles and sperm epididymis. This study supports the view that the stressor stimulus could be involved in the infertility process.   

  1. Conclusions (394-399 lines)

According to our results, it could be concluded that CVS damage triggers the induction of apoptosis markers by intrinsic (PPAR-γ, p53, p21, HIF-α, and Bax) and extrinsic (p-Akt, AP-2α, and FAS) caspase-3 dependent pathways in both testicles and sperm epididymis. This study supports the view that the stressor stimulus could be involved in the infertility process, and future studies should identify the role of the antioxidant/oxidative stress balance in apoptotic gene expression. 

  • Observation 3: The English could be improved to more clearly express the research.

Answer: The English from all manuscript was corrected to more clearly express the research.

Reviewer 2 Report

Comments and Suggestions for Authors

In my opinion, the authors did not sufficiently revise their MS and I did not receive comprehensive answers to my questions either in the revised MS or in the covered letter.

Major points

  1. I still believe that the authors should not talk about increased apoptosis in testicular and epididymal cells based on the data presented, since they actually reported increased expression and activity of pro-apoptotic factors, not apoptosis itself. The authors also report that the sperm analysis was described in a previous paper by their group and provide a link to the master thesis. I found that the link is invalid, so it is impossible to read this paper. Why did the authors not include these unpublished data in the MS?
  2. The authors detailed the procedure for collecting epididymal but not the testis material. It was the latter material that was in the focus of my question about possible contamination by cells other than sperm. I still cannot be sure that the data presented relate specifically to sperm and not to other testicular cells, such as spermatids or even interstitial cells.
  3. The authors described the CVS procedure in sufficient detail and explained the need for its use. However, the details of the experiment remained unclear. The authors write: “eight rats were placed in a collective cage usually designed for four to five rats.” How could eight rats be placed in a collective cage if there were only seven animals in each group? The authors then write: “five rats were placed together in a plastic tank.” If there were seven rats in the group, what did the remaining two rats do?
  4. As I noted earlier, the content of malondialdehyde in the testis, as can be seen in Fig. 1, are almost twice as high for the control group as for the experimental group, indicating a reduction in oxidative stress during chronic stress exposure. Unfortunately, the authors did not provide a reasoned explanation for this interesting observation. The text fragment they added to the Discussion section is largely speculative and contains only general phrases about the mechanisms and factors of oxidative stress.
  5. Taking into account my comments made earlier, I believe that the Conclusion section as it currently stands is too general and superficial and should be completely rewritten.

Author Response

Dear Editor and reviewers

Thank you very much for taking the time to review this manuscript. Please find the detailed responses below and the corresponding corrections highlighted in red in the re-submitted file:

Reviewer 2

  1. I still believe that the authors should not talk about increased apoptosis in testicular and epididymal cells based on the data presented, since they actually reported increased expression and activity of pro-apoptotic factors, not apoptosis itself. The authors also report that the sperm analysis was described in a previous paper by their group and provide a link to the master thesis. I found that the link is invalid, so it is impossible to read this paper. Why did the authors not include these unpublished data in the MS?

Answer: Thank you for the observations, the rhetoric throughout the manuscript was changed and only changes in the expression of markers related to apoptosis are mentioned. The link to open Nolasco Garduño's master thesis was incorrect, it has now been corrected (see reference) and can be consulted in the repository, along with the fact that the research group is preparing a manuscript with histology, sperm quality, immune status, with this unpublished data.

Additionally, the following information was added in the results section:

Results (258-262 lines):

The sperm quality was reported previously in unpublished data from the master thesis of Nolasco Garduño [24] and was corroborated in this study. All the parameters decreased in the CVS group with progressive motility (t= 2.237, p=0.0378), viability (t=2.569, p=0.0249), concentration (t=2.613, p=0.0225), and morphology (t= 7.54, p=0.0001) compared with the control group.

as well as in the discussion (342-344 lines):

In this study, CVS triggers the induction of apoptosis markers involved in the intrinsic and extrinsic pathways; this is consistent with the decrease in viability and motility in the sperm of the CVS group, previously reported by Nolasco Garduño [24].

The correct link is: http://repositorio.uatx.mx:8443/jspui/handle/DSyTI_UATx/201

  • Observation 2: The authors detailed the procedure for collecting epididymal but not the testis material. It was the latter material that was in the focus of my question about possible contamination by cells other than sperm. I still cannot be sure that the data presented relate specifically to sperm and not to other testicular cells, such as spermatids or even interstitial cells.
  • Answer: Thank you for the observation, you are right, for the testicle experiments, the complete organ was used and homogenized, this has been noted throughout the manuscript and the corrections in the methodology, results and discussion have already been made in red.

In the methodology section was added the following information:

(116-123 lines):   The rat was placed in a supine position and a longitudinal incision was made along the ventral midline of the abdominal wall to extract the complete testicles and weigh them on a scale (ACCURIS W3300-500). Both testicles were frozen in an ultra-freezer at -80°C for subsequent analysis. The sperm epididymis was taken from the middle to the cauda of the epididymis, with a single incision, and placed in an Eppendorf tube containing 2 mL of phosphate-buffered saline (PBS). A tissue sample from the cauda epididymis (300 mg), or complete testicle was taken and placed in an Eppendorf tube containing 2 mL of phosphate-buffered saline (PBS)

  • Observation 3: The authors described the CVS procedure in sufficient detail and explained the need for its use. However, the details of the experiment remained unclear. The authors write: “eight rats were placed in a collective cage usually designed for four to five rats.” How could eight rats be placed in a collective cage if there were only seven animals in each group? The authors then write: “five rats were placed together in a plastic tank.” If there were seven rats in the group, what did the remaining two rats do?
  •  

Answer: Thank you for the observation, the inconsistencies in the methodology related to the CVS have been corrected in the methodology, which is as follows:

2.2. Chronic Variable Stress procedure model

The rats were subjected to a sequence of alternative and random individual stressors each day at different times to each rat. The following stressors were used: (a) Space reduction in clean box stimuli, the seven rats were placed in a collective clean box designed for four to five rats, with dimensions 50 cm x 40 cm x 21 cm for 5 h; (b) space reduction in dirty box stimuli, the seven rats were placed in a collective dirty box designed for four to five rats, with dimensions 50 cm x 40 cm x 21 cm for 5 h; (c) restraint stimuli, for 3 h each rat, one by one, were placed into a plastic tube of 6 cm x 20 cm avoiding movement and risk of tipping over; (d) forced swimming stimuli in cold water, each rat, one by one, were placed together in a plastic tank, dimensions 100 cm height x 50cm diameter containing 75 cm of water at 18°C for 10 min in cold water ; (e) forced swimming stimuli in warm water; same procedure before at 28°C for 10 min. Animals remained on their diets throughout stress exposure. Stress application started at different times every day to minimize its predictability (23).

  • Observation 4: As I noted earlier, the content of malondialdehyde in the testis, as can be seen in Fig. 1, are almost twice as high for the control group as for the experimental group, indicating a reduction in oxidative stress during chronic stress exposure. Unfortunately, the authors did not provide a reasoned explanation for this interesting observation. The text fragment they added to the Discussion section is largely speculative and contains only general phrases about the mechanisms and factors of oxidative stress.
  •  

Answer:

Thank you for the observation, it is an interesting and difficult question to answer, we will do our best to explain it.

 In this study, the testicle of CVS rats decreases the MDA levels almost 50% than control group. In other studies, has been reported that stress stimuli could be increase the levels of dihydrotestosterone (DHT), a biologically active metabolite of the hormone testosterone; it is formed in the prostate gland and testicles by the enzyme 5α-reductase, this hormone increasing by exercise-related stress (Sgro P 2021); Also, previous studies have demonstrated the antioxidant power of DHT in response to allostatic changes and adaptative mechanism to mitigate the damage induced by stress stimuli and to prevent the formation of lipid peroxides (LPO) (Na Lee M, 2008), this could explain in part a MDA reduction in our study; however, it is necessary in future investigation to know the DHT, and antioxidant enzymes levels, and corroborating the DNA damage in testicle and sperm epididymis.

This information was added in the discussion section in red as follow:

These suggest that the balance of glucocorticoids and androgens could be related to the antioxidant/oxidant status in the testis. In addition, this study shows a clear trend of 50% lower MDA levels in the CVS group than in the control group. In other studies, it has been reported that stress stimuli could increase the levels of dihydrotestosterone (DHT), a biologically active metabolite of the hormone testosterone; it is formed in the prostate gland and testicles by the enzyme 5α-reductase, this hormone increasing by exercise-related stress [30]. Also, previous studies have demonstrated the antioxidant power of DHT in response to allostatic changes and adaptative mechanisms to mitigate the damage induced by stress stimuli and to prevent the formation of lipid peroxides (LPO) [31]. This could explain in part the MDA reduction in testicle as a response to CVS in our study; however, it is necessary in future investigation to know the DHT levels and the antioxidant enzyme levels. Related to epididymis, it is equipped with a battery of antioxidant enzymes that maintain the ROS at physiological levels to achieve the goal of protecting the maturing spermatozoa [32]. In support, previous data in our group showed a decrease in SOD, CAT, and total GSH, suggesting that reactive species generated during stress could be neutralized [24]. In the sperm epididymis, CVS induces an imbalance between oxidative stress and antioxidant status, suggesting that deregulation of the redox state causes damage to the sperm epididymis. Different studies report that stress stimuli lead to decreased SOD or CAT activities and diminish total GSH [11]. It has been previously reported that CVS decreases the activity of antioxidant enzymes associated with an imbalance in the redox state [33].

Observation 5: Taking into account my comments made earlier, I believe that the Conclusion section as it currently stands is too general and superficial and should be completely rewritten.

Answer: thanks for the observations, the conclusion was modified in both sections as follows: 

Abstract:    (29-32 lines)  In conclusion, CVS damage triggers the induction of apoptosis markers by intrinsic (PPAR-γ, p53, p21, HIF-α, and Bax) and extrinsic (p-Akt, AP-2α, and FAS) caspase-3 dependent pathways in both testicles and sperm epididymis. This study supports the view that the stressor stimulus could be involved in the infertility process.  

  1. Conclusions (394-399 lines)

According to our results, it could be concluded that CVS damage triggers the induction of apoptosis markers by intrinsic (PPAR-γ, p53, p21, HIF-α, and Bax) and extrinsic (p-Akt, AP-2α, and FAS) caspase-3 dependent pathways in both testicles and sperm epididymis. This study supports the view that the stressor stimulus could be involved in the infertility process, and future studies should identify the role of the antioxidant/oxidative stress balance in apoptotic gene expression. 

  • Observation 6: The English could be improved to more clearly express the research.

Answer: The English from all manuscript was corrected to more clearly express the research.
